# The Submissive Relationship of Public Health to Government, Politics, and Economics: How Global Health Diplomacy and Engaged Followership Compromise Humanitarian Relief

**DOI:** 10.3390/ijerph17041420

**Published:** 2020-02-22

**Authors:** Daniel Peplow, Sarah Augustine

**Affiliations:** 1Department of Health Services, University of Washington, White Swan, WA 98952, USA; 2Suriname Indigenous Health Fund, White Swan, WA 98952, USA; saugustine08@gmail.com

**Keywords:** economic development, community health, structural causes, engaged followers

## Abstract

This paper describes efforts by public health practitioners to address a health crisis caused by economic development policies that are unrestrained by either environmental, public health, or human rights mandates. Economic development projects funded by international funding institutions like the Inter-American Development Bank that reduce poverty when measured in terms of Gross Domestic Product (GDP) per capita in the transborder region between Suriname and French Guiana harm minority populations where commercial activities destroy, alter, and remove the resources upon which local communities depend. In this study, the structural causes of a community health crisis affecting Indigenous people in the transborder region between Suriname and French Guiana was addressed by seeking gatekeepers in government who have access to policy-making processes. We found that deeply rooted economic development policies structured social, economic, and political alliances and made them resistant to feedback and reform. We concluded that work must be focused beyond the simple exchange of public health information. Public health practitioners must become politically active to create new policy commitments and new patterns of governance that advance development as well as improve health outcomes. Failure to do so may result in public health practitioners becoming ‘engaged followers’ that are complicit in the inhumanity that springs from their acquiescence to the authority of government officials when their policies are the cause of preventable death, disease, and disability.

## 1. Introduction

The colonial period in the transborder region between Suriname, French Guiana, and northern Brazil lasted approximately 300 years from around 1650, the end of the period when Indigenous people populated the entire region, to the end of the plantation economy in 1950. After World War II, the decolonization process was accompanied by foreign aid, also referred to as development aid, development assistance, concessional flows, or simply aid. This aid dominated the economic, cultural, and political relationship between the industrial world and the developing world. By the mid 1960s, most aid obtained by the developing world was in the form of loans and not grants. The outcome was that the development process was inevitably accompanied by indebtedness [1].

The region surrounding the Maroni River (Figure 1), which forms the border between Suriname and French Guiana, is also a contact zone where distinct social entities meet, clash, and cooperate with each other under highly asymmetrical relations of power. Instead of a border between two postcolonial countries, the Maroni River is a place of intense interaction between native Amazonians, tribal descendants of escaped African slaves, international investors, development agencies, and multinational enterprises (MNEs). Under the economic pressure of debts from foreign development loans, governments in Suriname and French Guiana became reliant on commodity exports and were committed to macroeconomic open market operations [2].

The decline of the plantation economy meant regional governments had a pressing need for new sources of State income to support the decolonization process, balance payments, and meet international loan obligations. Gold mining companies and expanding operations by MNEs have replaced plantations as the major players in the post-colonial economy [3]. The nature of gold mining means operators move throughout the traditional areas of Indigenous communities competing for natural resources [4]. State and international legislation grant mining rights to mine operators on land traditionally held by Indigenous and tribal peoples [5].

Since WWII, increased national wealth, as measured in terms of gross domestic product (GDP), has improved the lives of millions of people in developing nations [6]. Since 1990, the global maternal and infant mortality rates have been cut in half. However, in places in the developing world experiencing major social disruption, more than a thousand women die every day from preventable causes during pregnancy and childbirth, including the Guiana region [7].

In the Guiana region, Peplow and Augustine showed that the areas in which the tribal and Indigenous communities reside are heavily contaminated with mercury used by miners to process gold. Mercury exposure was found at levels causing adverse neurological effects among people in the Guiana region who are highly dependent on fish in their diet [8,9]. Health assessment studies showed victims are likely to have their physical and cognitive abilities damaged irreversibly [10]. There are reports that suicide and attempted suicide among the Indigenous populations of the upper Maroni occurred at a rate 13 times higher than the regional average [11].

Martin et. al. propose that these preventable deaths, diseases, disabilities, and suicides occur due to a problem of collective decision making rather than a problem of scarce resources [7]. Their work showed clear evidence that investing in health improvements also contributes to significant increases in GDP and failure to invest in heath represents an important threat to global prosperity.

According to Banerjee and Duflo, failure of economic models to prioritize a thriving and healthy environment is what economists call “misallocation”. Whereas misallocation saps growth, only reallocation can improve it [6]. As global health practitioners, we persist in bringing the voices of the excluded and affected peoples into the decision-making process.

A basic premise of our work is that the governance response to the global health crisis in the Guianas is determined less by the compelling logic of an appropriate response, and more by a contest between competing interests and the operation of power. One explanation is that truth, as it is perceived by people in a democracy, varies between interest groups and is different than the empirical truth found in a global health research notebook. Politicians and policy makers perceive truth based on a contest between competing interest groups and not on empirical data.

This is a classic problem where public health practitioners find themselves working at a crossroads that leads in two directions. One pathway addresses the social, political, and economic foundations of health at the system level, and the other pathway a clinical approach that is more narrowly focused on proximal risk factors that are technical in nature [12,13]. Among the Indigenous people there are factors controlling health and well-being that lie outside the health sector and are socially and economically formed. This reality suggests that our fundamental attention in public health policy and prevention should not be directed solely towards a search for technical or behavioral solutions to health problems at the individual level, but also toward breaking down existing social, political, and policy barriers to minimize the root causes of disease, disability, and premature death.

Machenbach cautioned against an emphatically political approach to public health because, in the long run, it could prove to be a self-defeating strategy, and because of the dangers of politicization [14]. Politics is divisive, and long-term support for global health practitioners and their initiatives can be eroded as well as strengthened by recurrent political debates. Machenbach proposed an imaginary “ladder of political activism” with four rungs from which the global health practitioner can choose the level of political action they think would be most effective and appropriate (Figure 2). The first rung is that of political passivism where information on health risks and opportunities for health improvement is exchanged within the health sector only, and politicians are only informed if they ask for information. On the second rung, public health professionals actively disseminate relevant information among politicians by addressing their reports to the government. At the third rung, public health professionals try to directly influence the political process by lobbying and by actively engaging politicians of specific political parties. On the highest or fourth rung, public health professionals become politicians themselves, and try to obtain positions in government or parliament to reach their objectives.

The goal of this study was to test whether the input from an academic researcher and global health practitioner working at the community level could join government (public) and industry (private) as partners in a public–private partnership (PPP). The objective of this study was to access staff, hearings, meetings, and advisory panels to contribute to the resolution of complex problems caused by factors that lie outside the health sector and are socially and economically based [15].

Inherent in this objective were three critical assumptions. The first assumption was that public health practitioners, academics, researcher-analysts, and leaders at the community level, and other members of Civil Society, could join government (public) and industry (private) as partners in PPPs to share the risks, responsibilities, and decision-making processes with the objective of collectively and more effectively addressing a common goal [16]. The second assumption was that communication channels between those inside and those outside of government are open and ideas and information pass through these channels in a network of involved people, somewhat independent of their formal positions. The third assumption was that common values, orientations, and world views would form bridges between those inside and those outside of government.

## 2. Methods

This project examined the government sector for areas where public health practitioners could participate in joined-up leadership within government, across sectors, and between levels of government as called for in the World Health Organization’s (WHO)’s statement on Health in All Policies [15]. While we understood our own roles as public health practitioners well, we had only a rudimentary and fragmentary understanding of the agenda building and decision-making process in government, including where the boundaries between the public and private sector were. These conditions made our participation in the policy-making process fluid. Consequently, we operated a lot by trial and error, by learning from experience, and by pragmatic invention.

We addressed the causes of health and well-being that lie outside the health sector, which were socially and economically formed, using a three-step policy-making model framework: (1) there is a problem that violates universally shared values; (2) civil society transmits information about the problem to government; (3) after steps 1 and 2 have been completed, the necessary and appropriate laws or decisions would be made by the government to resolve the problem [17].

To guide our work and bring structure to our participation in the policy decision-making process, we used the Multiple Stream model that divided the policy process into three ‘streams’: the problem stream, the policy stream, and the political stream [18,19]. This report focuses on the problem and policy stream only. This report does not address the political stream or processes that revolve around the consideration of elections, and partisan forces. The problem identification and problem-solving streams revolved around the processes used to set the political agenda. Whereas, the policy stream focuses on processes that put problems on the political agenda so they can be turned into concrete solutions.

This report does not distinguish between the governmental agenda, which is the list of subjects that are getting attention and the decision agenda, which is the list of subjects within the governmental agenda that are up for an active decision. We made a distinction between the visible specialists (e.g., elected officials, high-level appointees, political parties, and campaign), who we did not engage directly, and the hidden specialists (e.g., academics, career bureaucrats, congressional staffers, and corporate executives) who we sought to engage.

This report also refers to Johan Mackenbach’s Imaginary Ladder of Political Activism as a conceptual framework for engaging ‘gatekeepers’ (staffers, at hearings and meetings, and among participants on advisory panels) at all levels of government (local, regional, national, and international) [14]. We moved beyond the first rung of Machenbach’s ladder, referred to as political passivism, where information on health risks and opportunities for health improvement would be exchanged within the health sector only as peer reviewed publications. Politicians and presumably the communities affected would be informed if they asked for it.

The second rung of Mackenbach’s Imaginary Ladder of Political Activism was chosen because it describes efforts to actively disseminate relevant information to gatekeepers. No attempt was made to use the third rung of political activism to attempt to directly influence the political process by lobbying and by actively engaging politicians of specific political parties was made. Additionally, no attempts were made to work on the fourth rung where public health professionals become politicians themselves and try to obtain positions in government to reach their objectives.

The initial approach for each contact was made through an emailed letter of interest that included indicators, focusing events, and feedback (described below) to show that there is a problem worth consideration as described by Kingdon [19]. Gatekeepers, including staff persons, appointees, career bureaucrats, and nongovernmental actors, were selected to call attention to the structural causes of public health problems and possible government solutions. All emailed letters of interest concluded with a statement of interest to contribute to and participate in, as a representative of the public health sector, high level policy processes to address the causes of health and well-being that lie outside the health sector and are socially and economically formed. The indicators, focusing events, and feedback statements were used as a basis for identifying common values, orientations, and world views; to form bridges between the public health practitioner and gatekeepers; and to engage civil society as a third party in a PPP.

Ethics Approval: the primary intent of the work discussed in this paper was aimed at a specific public health problem, specifically to support indigenous people in the Guiana region with their efforts to self-diagnose public and environmental health problems concomitant with economic development projects and exposure to Hg contamination from mining. This paper addresses the secondary benefits of the community-led efforts at the interface between health, well-being, and economic development. These public health intervention projects were reviewed by the University of Washington Institutional Review Board (IRB), Human Subjects Division (HSD). No reference number was assigned because the community-led project was conducted as a public health service and was deemed “non-research”. Consequently, the work was not considered to be within the purview of institutional review. As such, these non-research investigations have yielded insights of generalizable value that merit dissemination, but the research versus non-research determination, which is based on the primary intent, remains unchanged.

### 2.1. Indicators

The indicators used to show that there is a problem worth consideration were (1) 58% of previously analyzed hair samples showed the Indigenous communities in question had mercury levels above the World Health Organization safety limit (10 ppm), and (2) case studies based on a battery of neurological tests in conjunction with hair mercury data found clinical evidence of mercury poisoning among Indigenous Wayana people living in the transborder region between Suriname and French Guiana [10].

### 2.2. Focusing Events

Two *focusing events,* also referred to as powerful symbols [19], were used call attention to the problem: (1) the documented annual release of 44–87 tonnes of mercury from small-scale gold mines in Suriname and French Guiana annually [20,21] ranks as an environmental disaster comparable to other global disasters such as the Minamata Disaster in Japan (mercury contamination in Japan in the 1950s) [22], the Fukushima Daiichi Nuclear Power Plant (FDNPP) accident [23], Thalidomide scandal [24] (pharmaceutical birth defects in the 1960s), and the Fukushima Daiichi disaster (nuclear meltdown in 2011); and (2) the continued monitoring by public health researchers of impacted people who have endured decades of research without benefit, especially pregnant women and newborn children exposed to mercury, is comparable to the notorious United States Public Health Service’s untreated syphilis study in Tuskegee (1932–1972) and its sexually transmitted diseases inoculation research studies in Guatemala (1946–1948) [25], or to the more recent Baltimore lead-paint study from the 1990s carried out in Baltimore and overseen by Johns Hopkins University [26]. Entire communities, races, and ethnic groups are at risk while policies and procedures have already been developed, but not implemented, to prevent these outcomes.

### 2.3. Feedback

We offered informed feedback to government officials about the operation of existing programs designed to address the problems identified above: (1) the World Health Organization’s Adelaide Statement on Health in All Policies [15], (2) the IADB’s (Inter-American Development Bank) Strategy and Operational Policy for Indigenous Development [27], and (3) United Nations Security Council Responsibility to Protect Resolution [28].

The level of importance of our petition to staff, bureaucrats, participants on advisory panels, meetings, hearings, interest groups, administration officials, congressional staffers, and administration appointees to the request for joined-up leadership within government, across sectors, and between levels of government were scored using an Index of Importance according to the criteria:INDEX OF IMPORTANCE (Total Score 0)—global health agenda is set by actors outside the academic researcher-analyst community who:
1.1.*Did not acknowledge* receipt of literature and reports submitted directly by the academic and researcher-analyst community and leaders at the community level; Score = 01.2.*Would not discuss* the literature and reports submitted directly by the academic and researcher-analyst community and leaders at the community level; Score = 01.3.*Would not facilitate* academic and researcher-analyst community and leaders at the community level access (direct or indirect) to hearings, meetings, and advisory panels. Score = 0INDEX OF IMPORTANCE (Total Score 1)—global health agenda is set by actors outside the academic researcher-analyst community who:
2.1.*Acknowledged* receipt of literature and reports submitted directly by the academic and researcher-analyst community and leaders at the community level; Score = 12.2.*Would not discuss* the literature and reports submitted directly by the academic and researcher-analyst community and leaders at the community level; Score = 02.3.*Would not facilitate* academic and researcher-analyst community and leaders at the community level access (direct or indirect) to hearings, meetings, and advisory panels. Score = 0INDEX OF IMPORTANCE (Total Score 2)—global health agenda is set by actors outside the academic researcher-analyst community who:
3.1.*Acknowledged* receipt of literature and reports submitted directly by the academic and researcher-analyst community and leaders at the community level; Score = 13.2.*Discussed* the literature and reports submitted directly by the academic and researcher-analyst community and leaders at the community level; Score = 13.3.*Would not facilitate* academic and researcher-analyst community and leaders at the community level access (direct or indirect) to hearings, meetings, and advisory panels. Score = 0INDEX OF IMPORTANCE (Total Score 3)—global health agenda is set by actors outside the academic researcher-analyst community who:
4.1.*Acknowledged* receipt of literature and reports submitted directly by the academic and researcher-analyst community and leaders at the community level; Score = 14.2.*Discussed* the literature and reports submitted directly by the academic and researcher-analyst community and leaders at the community level; Score = 14.3.*Facilitated* academic and researcher-analyst community and leaders at the community level access (direct or indirect) to hearings, meetings, and advisory panels. Score = 1

### 2.4. Criteria

It was decided that success would be the identification of at least one ‘gatekeeper’ who would facilitate the introduction of the indicators, focusing events, and feedback information (Total Index of Importance Score of 3) to the appropriate hearings, meetings, and advisory panels [19]. We did not decide on a point at which we would consider our search a failure and stop the program.

Inherent in the objective of this project were three critical assumptions: the first assumption was that public health practitioners, academics, researcher-analysts, and leaders at the community level, and other members of Civil Society, could join government (public) and industry (private) as partners in PPPs to share the risks, responsibilities, and decision-making processes with the objective of collectively and more effectively addressing a common goal.

The second assumption was that communication channels between those inside and those outside of government are open and ideas and information pass through these channels in a network of involved people, somewhat independent of their formal positions.

The third assumption was that common values, orientations, and world views would form bridges between those inside and those outside of government.

## 3. Results and Discussion

The indicators, focusing events and feedback, as described in Section 2 above were sent to a total of 37 potential gatekeepers in a network of public–private partnerships including elected officials and other people, private or public, with active roles inside government. Fourteen candidate gatekeepers ostensibly agreed to meet to discuss our proposed global health agenda. Only five discussed our proposed community health agenda and met the criteria for Index of Importance Level 3: (1) the Inter-American Development Bank’s (IADB), (2) Independent Consultation and Investigation Mechanism (ICIM), (3) the International Committee of the Red Cross (ICRC), (4) the U.S. Department of Justice (DOJ), and (5) Corporate Social Responsibility (CSR) executives from a mining company. The other nine considered our petition to be of little importance.

### 3.1. Gatekeepers Scoring—Index of Importance Level 3

#### 3.1.1. Inter-American Development Bank (IADB)

When asked to engage in dialogue and consider the full implementation of the IADB’s Operational Policy on Indigenous Peoples and Strategy for Indigenous Development (IADB) to prevent lending projects from inflicting serious health and human rights abuses on local communities, an operations specialist for the IADB responded by saying that the natural resource sector is driving the economy in the Greenbelt region of Suriname and French Guiana and that the Suriname Land Management Project (SLMP), presented to the government of Suriname in 2006, was the final solution that would settle all land disputes in Suriname, including Indigenous lands in the southern interior region where gold and timber resources are concentrated [29].

According to the IADB, with the exception of those areas experiencing major social disruption from the reallocation of underutilized natural resources, economic development since WWII has been remarkably good for the world’s poor [30]. The IADB did not accept the premise of our proposal, which was that from a public health perspective the assimilation model inherent in Initiative for the Integration of the Regional Infrastructure of South America (IIRSA) and the SLMP, which are development projects linking South America’s economies to North American and Europe through new transportation, energy, and telecommunications projects, are examples of projects causing major social disruptions from a public health perspective [8,9,10]. The IADB disavowed the risk of harm to the health of Indigenous people by IADB financed economic development projects. Structural harm was a condition described by Kingdon to be a creation of the marketplace that responds to the demands of wealth and not to the basic needs of Indigenous communities undergoing involuntary assimilation. Furthermore, the marketplace did not represent a set of “common values, an orientation, or a world view that would form bridges between affected individuals and the IADB” [19].

During our last meeting with the IADB in 2019, officials indicated that they were not aware that in 2019, an historic U.S. Supreme Court decision recently concluded that international financial organizations like the IADB could be sued in U.S. courts for lending projects that inflicted serious human rights abuses on local communities, and then left the communities to fend for themselves [31]. They requested more information from us regarding the decision.

#### 3.1.2. The Independent Consultation and Investigation Mechanism (ICIM)

A petition was presented to the ICIM, an autonomous organ of the IADB, listing the threats Indigenous communities are facing due to socioeconomic and governance conditions generated by development investment. The petition specifically highlighted the negative impacts on health due to gold-mining operations [32]. As an overall issue, the petition mentioned the improvements needed at the policy-level to address the issues and to improve the situation facing the Indigenous people. The petition asked for support to engage in dialogue with high-level stakeholders to generate a new model that prioritizes thriving and healthy communities and that guarantees that all citizens will experience a decent quality of life. The petition noted that the racial hostility Indigenous people suffer and their lack of opportunity to participate in the majority culture or benefit adequately from resource distribution prevents Indigenous communities from becoming permanent and legitimate components of the majority society.

It was our stated purpose to address the structural impediments to community health. Previous health studies conclusively showed neurologic dysfunction consistent with mercury poisoning [8,9,10]. In this context, the petition emphasized the importance of overcoming the social, political, economic and ethical determinants of health that are responsible for the public health crisis facing Indigenous communities that are caused by economic development and resource extraction projects. The petition asked for the assistance of the ICIM to determine whether action can be taken in either of two ways to address the community health situation: (1) reducing preventable diseases at the policy level, and (2) by providing immediate relief at the community level to stabilize the situation over the short-term. Addressing the first recommendation, our petition recommended the IADB’s Operational Policy on Indigenous Peoples and Strategy for Indigenous Development be fully implemented [27]. To address the second recommendation, the petition to the ICIM calls for an inclusive economic development process that would devolve power to the level of local communities and enable Indigenous communities to act collectively to improve their own lives [33,34,35,36,37].

In response, the ICIM invited our team to review their 2014 Revised Draft Policy of the Independent Consultation and Investigation Mechanism but did not address our two core recommendations: the first, that an objective be added to provide a mechanism and process independent of bank management in order to investigate allegations of harm produced by the bank’s failure to comply with its own relevant operational policies in bank-financed operations. The second, regarding technical exclusions, the ICIM should not exclude valid requests on technical grounds because they are filed more than 24 months after the last disbursement of a relevant bank-financed operation. Both recommendations were rejected. The ICIM did not promise help advancing our proposed agenda suggesting their agreement to meet was for purposes of discovery and reporting, not intervention.

#### 3.1.3. U.S. Department of Treasury (DOT)

During a meeting in 2015, we discussed the ‘shadow banking’ system, in which credit is intermediated through a securitization process that backs development loans from development banks and other international financial institutions (IFIs) with the natural resources of developing countries like Suriname. Similar mechanisms occur internally in French Guiana but are held privately and are not visible for evaluation. These secured funding mechanisms, including asset-backed commercial paper (CP), asset-backed securities (ABS), and collateralized debt obligations are then turned in to derivatives and traded on the open market. The lead U.S. Department of Treasury representative reminded us that, “You are in the office of the United States Government and you cannot come in here and talk about securitization. If you do you better have legal representation”.

#### 3.1.4. International Committee of the Red Cross (ICRC)

Economic development plans in the Americas follow the Assimilation Model, previously codified in the U.S. as the Indian Removal Act of 1830. This act had three phases: removal, acculturation, and disappearance. Following the assimilation model, the agencies executing IIRSA and the SLMP held that the Indigenous communities occupying land and controlling natural resources that were considered to be ‘under-utilized’ and not meeting the criteria for ‘highest and best use’ would be displaced and the issues related to Indigenous land tenure would be ‘resolved’ within two generations [29,38].

Our argument was that IADB-financed SLMP and IIRSA projects are causing a health crisis among Indigenous people who are forced to abandon their traditional lifestyle and compete for livelihoods and safe living environments in the market economy. As a consequence of the assimilation model being used, life expectancy in affected communities is reduced approximately 17 years [39]. This is a slow-moving crisis of death by attrition among marginalized communities living in extreme poverty. This, in turn, creates a fast-moving crisis on a global scale. The reason: physical security and political security go together. The health of minority populations is a requisite for sustained human development and national security. Those who feel insecure about their survival needs have a fundamentally different outlook and political behavior from those who feel secure.

Ethnic minority groups who face long-term repression by a strong State, who are forced to assimilate into the majority culture, and who are subject to in-group/out-group forces can be viewed through the lens of social identity theory [40]. They face two options: first, they can attempt to assimilate to simply reduce their apparent threat to the state and minimize the in-group/out-group distinction, or second, they can attempt assimilation to take advantage of the tactical advantages that assimilation offers them against the State. It enables greater mobilization against the State allowing minority groups to engage in anti-State operations ranging from peaceful protests to violent terrorism or insurgency, in a bid to reduce the repression they face.

In our proposal to the ICRC, we suggested that a relief effort focusing on the public health crisis could provide a common ground around which many disciplines could come together to mitigate the ‘war’, albeit below the threshold of armed conflict, that is being waged. Like economic development, the voice of public health is often heard as a force for the greater good. This was the basis for a request made to the ICRC to provide immediate relief at the community level. The objective would be to stabilize the situation over the short-term given the mission of the ICRC to direct and coordinate activities that protect the lives and dignity of victims of structural violence (“other situations of violence”) as well as armed conflict.

We asked the ICRC to clarify and explain whether the violence perpetrated against the Indigenous people in the Guyana Shield region meets the criteria for ‘other situations of violence’ and fall within the ICRC’s field of action. We were given an ICRC policy document that explained the scope and limitation of the ICRC [41]. Although repressive or discriminatory State policies like *IIRSA* or the *SLMP,* that are responsible for acts of collective violence that have emerged against a backdrop of globalization, can have consequences that are even more far-reaching than those of armed conflicts, the ICRC was careful to exclude certain forms of collective violence when the State uses extensive police forces, and armed forces, to maintain or restore internal economic order.

#### 3.1.5. Mining Corporate Social Responsibility (CSR)

For mining companies operating in the Guyana Shield, CSR is about balancing diverse demands of shareholders and affected communities on issues such as environmental protection, public health, and human rights as well as the need to make a profit and meet the macroeconomic development goals of the country where they are working. Company executives, operating in the Guyana Shield, spend millions of dollars annually on CSR and still public opinion of the sector is poor. Maintaining a license to operate is a constant challenge. Columbus Gold Corporation in French Guiana is facing resistance from numerous social organizations to the expansion of gold mining activities in the Guyana Shield region based mainly on issues related to land tenure, environmental and community health impacts, and on the lack of community engagement.

Our proposal was for CSR programs to engage with affected communities and support community-based initiatives to address food, health, and education needs was met by mine executives who argued against involving Indigenous people, social organizations, as well as global health practitioners and human rights advocates because the complexity of the issues is ‘not discernable’ to people outside of government and business looking in. Indigenous communities interpreted this as an explicit rejection of the public health proposal.

### 3.2. Gatekeepers Scoring—Index of Importance Level less than 3

Beyond the five gatekeepers discussed above, there were other specialists who initially acknowledged and were willing to discuss our proposed public health agenda. Fourteen other gatekeepers agreed to meet but limited their interactions to interrogatories (legal requests for further information) designed to learn facts while holding back information of their own, giving misinformation, or promising help that never materialized. As a negotiating strategy, Anghelus and Boncu refer to this technique as ‘sandbagging’ [42]. They described this approach to resolve conflict as a negotiating technique that is distributive in nature rather than integrative. We realized that our preference for the integrative method of negotiating was naive. Our integrative negotiation approach viewed the negotiation relationship as one that was based on cooperation, trust, and flexibility. Williams and Rushton [43], addressing this disjuncture, suggest that the gatekeeper’s approach was from a different perspective that emphasized a distributive negotiating relationship based on competition, power, and control [42,44].

### 3.3. Competing Discourses

One explanation for this disjuncture between health needs and governance is that what is perceived as truth by policy makers in government is different than the empirical truth found in research reported by public health practitioners. That is, politicians, policy makers, and economists perceive truth based on a contestation between competing discourses and not on empirical data alone from the health sector [45]. Consequently, the policy response to a community health problem is contingent on a range of factors, some of which are technical in nature and specific to the health issue and in other cases the policy response is determined by a process that involves the competition between special interests and the operation of power.

Williams and Rushton recommend viewing each of the discourses in public health as individually coherent since each is characterized by particular forms of language and logic with regards to health [41]. They suggest that the manner in which a public health agenda could be accomplished would be by conforming the language in which the health argument is couched within the dominant discourse. For example, the security discourse should treat health in terms of threats and defenses and use threat-defense language. Contending parties within economics would have to use the same language and draw on statistics, cost-benefit analyses, and calculations of efficiency.

This raises an ethical question: although health can be the basis for establishing a dialogue and bridging diplomatic barriers because they transcend traditional and more volatile and emotional concerns [46], what ethical dilemmas arise when community health advocates accept the limits and use the language of non-humanitarian actors in the humanitarian sphere? Will humanitarian issues take precedence or will humanitarian issues be used to support political and economic interests? Sarewitz describes how the free market, the cornerstone of Western economies, would harm the health of communities like those of the Indigenous people in the transborder region between Suriname and French Guiana because the creations of the marketplace respond to the demands of wealth and not to the basic needs of Indigenous communities undergoing involuntary assimilation [17].

### 3.4. Engaged Followership

Despite the fact that Milgram’s work on followership and obedience to authority [47] has been widely criticized, the gatekeepers contacted all indicated they were committed to and constrained by mandates, policies, and the policy-making process and were not free to provide the academic and researcher-analyst community and leaders at the community level access (direct or indirect) to high level policy hearings, meetings, and advisory panels. There are two key reasons why this evidence could be viewed as consistent with claims that harm-doing is a product of engaged followership [48]. The first is that during our conversations with gatekeepers they felt compelled to defend themselves against being responsible for the health crises we described. The second is that the gatekeeper’s accounts of their actions revolved around their mandates, highest and best use of resources, and policies that contribute to the greater good, all expressions of trust in the high-level policy making process. We interpret these as manifestations of shared identity and engaged followership.

This suggests that there are limits to how far public health practitioners should go when they acquiesce to government leadership responsible for non-humanitarian policies. When the actions of public health practitioners take on a subordinate role, they may not be benign and may actually enhance the power of the dominant discourse as a follower. This raises another question: when the authority of a non-humanitarian leader has the propensity to inflict harm on others, is that harm enabled if public health practitioners accept inhumane State policies and practices when they are based solely on the premise that they contribute economically to the greater good?

The subordinate role that public health has to the political and economic agenda highlights the top-down nature of the policy making process. It also contradicts Kingdon’s assertion that the flow of information through the policy making process is characterized by its ‘extraordinary looseness’, and that ideas are introduced without restriction. Instead, we observed there is a hierarchical system in which lower level gatekeepers control the flow of information up and down, through channels, and to and from superiors [19].

### 3.5. Agenda Building

The outside initiative model [49] accounts for the process through which issues arise in nongovernmental groups and are then expanded sufficiently to reach the public agenda. Where the number of potential public issues far exceeds the capabilities of decision-making institutions to process them. Redefinition of issues may be needed to overcome the apathy of the gatekeepers that public health practitioners seek to involve. Therefore, an extended ‘expansion’ process is likely to include several redefinitions of the issues. As public health practitioners we also recognized the pressure to redefine our proposal to accommodate the interests of the gatekeepers away from the original interest. To avoid this, Machenbach suggested a fourth rung in the Imaginary Ladder of Political Activism where public health professionals become politicians themselves and obtain positions in government to reach Health in All Policy objectives [15]. This conforms to the ‘mobilization’ model described by Cobb [49] that describes how issues initiated inside government achieve formal agenda status almost automatically when they are initiated by Civil Servants.

### 3.6. Responsibility to Protect

When public health advocacy fails to have health and well-being included as a key component of policy development and the State fails to protect its people, either through lack of ability or a lack of willingness, the responsibility to protect shifts to the broader international community [50]. Typically, the responsibility for humanitarian relief as it relates to community health and well-being involves three sets of actors with different skill-sets and different capabilities: governmental bodies, non-governmental organizations (NGOs), and militaries [51]. There is the potential that the relationship between NGOs and the other groups (government and military) can become contentious when NGOs strive to appear apolitical and neutral while government and military involvement in humanitarian relief is a political act, driven by strategic goals that includes global health diplomacy.

An example is the 2005 MedReady program deployed in Suriname by the US Army to provide short-term medical services to affected communities in Suriname’s interior region. Ostensibly, MedReady sought, as an objective, to save lives and alleviate suffering of a crisis-affected population. Its overarching objective, however, was to provide training, build small unit leaders, and work out logistical protocols to enhance security and the military capacity to access remote areas and control unrest among an entire population of people who had been exposed to toxic levels of mercury [8,9,10] and whose cognitive response to displacement and forced assimilation was feared might lead to armed conflict [38]. The idea that global health diplomacy led by State actors is not founded on the principles of neutrality and impartiality highlights an ethical challenge to public health practitioners.

## 4. Conclusions

(1) Public issues that are the result of economic development policies are inherently politicized and are not readily resolved through the simple exchange of public health information within the health sector through peer reviewed publications or the active dissemination of public health information among staffers, at hearings and meetings, and among participants on advisory panels. These tactics will not lead to the new social contract that is needed between the health sector and government to improve health outcomes as well as advance development.

(2) Taking an integrative approach to joined-up leadership between public health practice and governance that views a negotiating relationship based on cooperation, trust, and flexibility is not adequate. To be effective, the public health practitioner must be prepared to deal with a different perspective in government that emphasizes a distributive negotiating relationship based on competition, power, and control.

(3) When public health practitioners highlight their potential contribution to the resolution of complex problems across government, the politicians, policy makers, and economists perceive proposals from the public health sector as challenge to other competing discourses and a desire to exercise power in the policy-making process.

(4) Public health practitioners providing policy feedback and seeking legislative reform will encounter patterns of governance, including adaptations, expectations, incentives, and legislative agendas, that conform to policy commitments inherited from the past and not practices that respond to the conditions of the day. Policies become deeply rooted and can be resistant to feedback and reform.

(5) There are risks when public health practitioners acquiesce to the authority of non-humanitarian experts in government: they might unwittingly share responsibility for the inhumanity that springs, not necessarily from intentional malevolence, bigotry, or pathology, but rather from the mundane inclination to accept the decision of officials in authority, however unreasonable or brutal their decisions may be.

(6) Public health professionals should consider the third rung of Machenbach’s Imaginary Ladder of Political Activism. This tactic involves lobbying and actively engaging politicians of specific political parties. While public health professionals may be subject to restrictions on the amount of “lobbying” in which they or their organization can engage, there are many activities they can engage in that can help educate policymakers and influence policy development that are not considered “lobbying” and therefore are not subject to limitations. The fourth rung is another option in which public health professionals themselves obtain positions in government that would position them to create new policy commitments, create new patterns of governance, and include expectations and incentives that address health considerations in all high-level policy processes.

(7) While working on long-term goals to address the social determinants of health, public health practitioners should also address the immediate need for culturally relevant palliative care and provide relief for communities facing urgent health crises.

## Figures and Tables

**Figure 1 ijerph-17-01420-f001:**
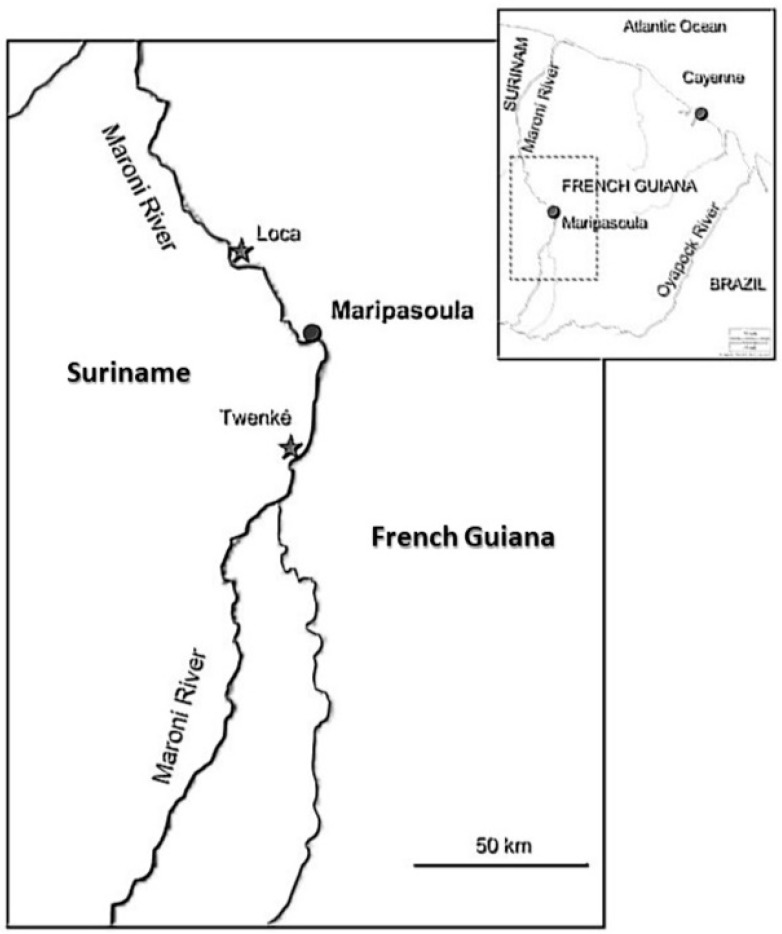
Map showing the transborder region between Suriname and French Guiana.

**Figure 2 ijerph-17-01420-f002:**
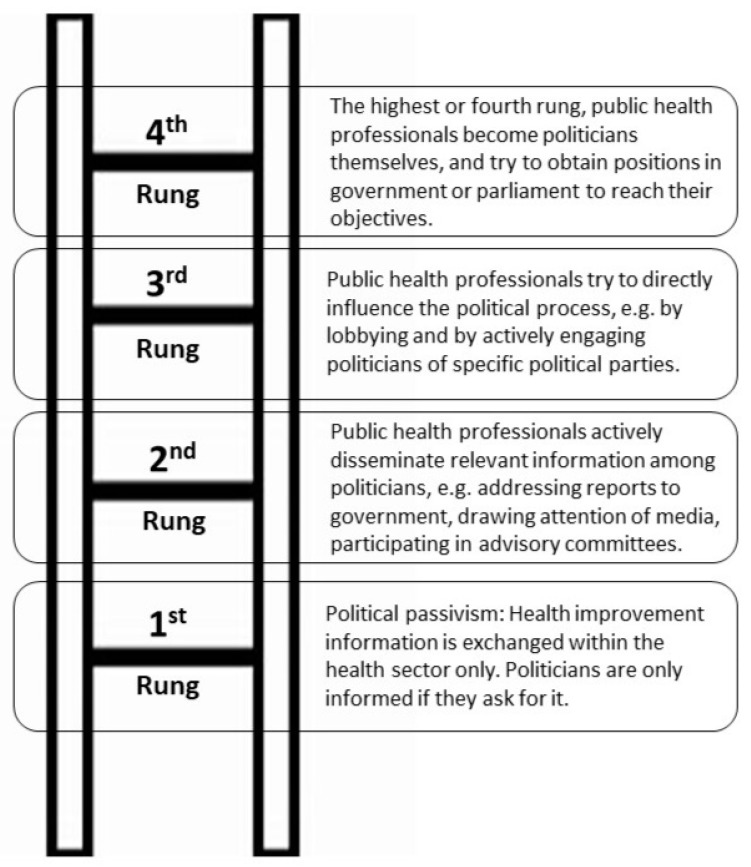
Machenbach’s “imaginary ladder of political activism” depicting four potential rungs of engagement available to public health professionals [14].

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
