# Peer review of "The Submissive Relationship of Public Health to Government, Politics, and Economics: How Global Health Diplomacy and Engaged Followership Compromise Humanitarian Relief"

_ijerph, 2020, doi:10.3390/ijerph17041420_

Round 1

Reviewer 1 Report

This article presents a case study of public health practioners working across levels of government, business and NGOs to promote the benefits of including pulbic health considerations in international development. Their efforts appeared ineffective, leading to the conclusion that political activism beyond provision of information is required to protect public health when conflicting interests are involved.

An original important and interesting article that should be published. However it requires re-working for clarity. I would like to see more background and history of the events being explored and better use of diagrams and tables to explain the findings, and improved sentence structure. 

Abstract: clear, concise, and represents the contents of the article

Introduction: Is there relevant history further back than the decade that this project describes? A history of colonisation may provide grounding for the project. I note that point 4 in the conclusion refers to the past and more historical background may assist in understanding the current issue.

Johan Mackenbach’s Imaginary Ladder of Political Activism sould be introduced in the introduction, rather than the methods. A diagram of the ladder would make it easier to understand.

A map would be useful for readers unfamiliar with the region.

Table is unclear. What do the headings “Not important, little importance, slight importance and very important” refer to?

Methods

The choice of two focusing events in this project (release of 44-87 tons of mercury and public health surveillance without intervention) should be in the introduction rather than the methods, with the reasons for selecting these.

The list of criteria for “not important, little importance, slightly important and very important” should be in a table rather than text.

Are these two lists on page 6 the same groups? Such long lists are difficult to conceptualise

List of page 6 line 96: elected officials, appointees, staff, bureaucrats, interest groups, lobbyists, consultants and  interest groups

List on page 6 line 52 staffers, bureaucrats, participants on advisory panels, meetings, hearing, interest groups, administration officials, congressional staffers and administration appointees.

These would be easier to read with some collective nouns, or else a table.

Results and Discussion

I found the structure of this confusing, perhaps because results and discussion were presented together. The headings seemed to partially correspond to the columns of Table 1.

It would be useful to know who were the other specialists referred to in the section headed “Other”. This "Other" section appeared to include discussion rather than results, including references to similar findings in the literature. It would be useful to have more background on the work of Williams and Rushton, Milgram, Machenbach, Cobb and Milgram to contextualise the discussion and better understand why the authors are referring to them.

Opening sentence on Inter-American Development Bank has 117 words, making it clumsy and dfifcult to understane. In general many of the sentences are so long that the meaning is lost.

Conclusion

Again, each point seems valid but the sentences are so long that the point is lost, particularly a problem in the conclusion that could provide sharp take away points. I felt that I agreed with the authors and could use the conclusion, but their passion is lost in the verbosity of the sentences.

Author Response

Please note. The manuscript was revised in two steps. First all comments were corrected as indicated below. Second, other corrections were made, paragraphs and sections were revised further which may have change the response and in some cases eliminated the text in question and the response.

1) Introduction: Is there relevant history further back than the decade that this project describes? A history of colonisation may provide grounding for the project. I note that point 4 in the conclusion refers to the past and more historical background may assist in understanding the current issue.

Authors: The history of colonialization and the plantation economy in the Guiana region is complicated. In summary, the colonial period could be seen to have lasted 300 years from 1650, the end of the period when indigenous people populated the entire region, to the end of the plantation economy in 1950. This and other revisions will be made to the Introduction and Discussion sections.

2) Johan Mackenbach’s Imaginary Ladder of Political Activismshould be introduced in the introduction, rather than the methods. A diagram of the ladder would make it easier to understand.

Authors: Diagram (Figure 2) created.

3) A map would be useful for readers unfamiliar with the region.

Authors: Map (Figure 1) created.

 4) Table is unclear. What do the headings “Not important, little importance, slight importance and very important” refer to?

Authors: Revised categories to an “Index of Importance”, assigned criteria to score index, replaced four original categories with categories that have index levels scored 0 – 3. Entire section (51-94) was revised to support and reflect changes to include Index of Importance made to Table 1, to clarify the categories, and to make them more quantitative.

5) Methods

The choice of two focusing events in this project (release of 44-87 tons of mercury and public health surveillance without intervention) should be in the introduction rather than the methods, with the reasons for selecting these.

Authors: The introduction now includes a statement that describes how “problems’ are not self-evident and that in addition to ‘indicators’ they need “a little push to get the attention of people in and around government, e.g., using focusing events. See Kingdon, JW Agendas, Alternatives, and Public Policies (2nd Edition), Longman, New York, pp 253.

6. The list of criteria for “not important, little importance, slightly important and very important” should be in a table rather than text.

Authors: Table revised to include “Index of Importance”, criteria to score index. Replaced four original categories in table and text with categories that have index levels scored 0 – 3. Entire section (51-94) was revised to support and reflect changes to include Index of Importance made to Table 1, to clarify the categories,  and to make them more quantitative.

7) Are these two lists on page 6 the same groups? Such long lists are difficult to conceptualise. List of page 6 line 96: elected officials, appointees, staff, bureaucrats, interest groups, lobbyists, consultants and  interest groups. List on page 6 line 52 staffers, bureaucrats, participants on advisory panels, meetings, hearing, interest groups, administration officials, congressional staffers and administration appointees.These would be easier to read with some collective nouns, or else a table.

Authors: Lists have been replaced with, “elected officials and other people, private or public, with active roles inside government”.

8) Results and Discussion. I found the structure of this confusing, perhaps because results and discussion were presented together. The headings seemed to partially correspond to the columns of Table 1.

Authors: Introduction and Discussion rewritten to make them clearer and more concise.

9) It would be useful to know who were the other specialists referred to in the section headed “Other”. This "Other" section appeared to include discussion rather than results, including references to similar findings in the literature. It would be useful to have more background on the work of Williams and Rushton, Milgram, Machenbach, Cobb and Milgram to contextualise the discussion and better understand why the authors are referring to them.

Authors: Section was rewritten.

10)Opening sentence on Inter-American Development Bank has 117 words, making it clumsy and difficult to understand. In general many of the sentences are so long that the meaning is lost.

Authors: Section was rewritten.

11) Again, each point seems valid but the sentences are so long that the point is lost, particularly a problem in the conclusion that could provide sharp take away points. I felt that I agreed with the authors and could use the conclusion, but their passion is lost in the verbosity of the sentences.

Authors: Section was rewritten.

Reviewer 2 Report

This article does not constitute a scientific work. Even less, if you consider the scope of the paper it was submitted to. This sentence in the abstract: "Public health practitioners must become politically active to create new policy commitments and new patterns of governance that advance development as well as improve health outcomes" states clearly, what this text is all about. Nothing more that neo-Marxist post-modernist propaganda and ideology not founded in any evidence whatsoever.

Author Response

Please note. The manuscript was revised in two steps. First all comments were corrected as indicated below. Second, other corrections were made, paragraphs and sections were revised further which may have change the response and in some cases eliminated the text in question and the response.

1. This article does not constitute a scientific work. Even less, if you consider the scope of the paper it was submitted to.

Authors response: Pg8, Lines 189 – 195 were deleted, “The politics of indigenous assimilation is like the politics of war. It is the politics of purposeful acts of using force to displace a People without compensation and eliminate Indigenous People’s hold on land and resources. The deeds of the international investment community are not unknown to development experts which is made evident in the Inter-American Development Bank (IADB) 2006 Report on Operational Policies and Strategies for Indigenous Peoples [29]. That this policy is allowed to lie dormant is an example of how bureaucracy works and the banality of health inequity when it is normalized by society.”

2. This sentence in the abstract: "Public health practitioners must become politically active to create new policy commitments and new patterns of governance that advance development as well as improve health outcomes" states clearly, what this text is all about. Nothing more that neo-Marxist post-modernist propaganda and ideology not founded in any evidence whatsoever. 

Authors response: After interest groups, the collection of academics, researchers, and consultants is the next most important set of nongovernmental actors. Their impacts vary in a number of important ways. They affect alternatives more than agendas. Studies show that agendas are set by forces and actors outside the researcher-analyst community. Politicians turn to the researcher-analyst for proposals that are relevant to their concerns. The specific nature of a politician’s concerns, therefore, will reflect the individual politician’s interests. If the politician is a public health professional interested in health equity then they will use the power of their office to create those new patterns of government. See Kingdon, JW Agendas, Alternatives, and Public Policies (2nd Edition), Longman, New York, pp 253.

3. Machenbach made note that “an emphatically political approach to public health may also in the long run prove to be a self-defeating strategy, because of the dangers of politicization. Politics is divisive, and long-term support for public health can be eroded as well as strengthened by recurrent political debates” which is manifest in your comments. Machenbach, J., Politics is Nothing but Medicine at a Larger Scale: Reflections on Public Health's Biggest Idea. Journal of epidemiology and community health, 2009 vol. 63, no. 3, pp 181-4.

Reviewer 3 Report

This is a topical issue that is applicable across the world.

However not all the conclusions are supported by the results. For instance, how are you sure that when public health professionals become politicians, they will create a new pattern of government? 

Author Response

Please note. The manuscript was revised in two steps. First all comments were corrected as indicated below. Second, other corrections were made, paragraphs and sections were revised further which may have change the response and in some cases eliminated the text in question and the response.

However not all the conclusions are supported by the results.

Authors response: Pg8, Lines 189 – 195 were deleted, “The politics of indigenous assimilation is like the politics of war. It is the politics of purposeful acts of using force to displace a People without compensation and eliminate Indigenous People’s hold on land and resources. The deeds of the international investment community are not unknown to development experts which is made evident in the Inter-American Development Bank (IADB) 2006 Report on Operational Policies and Strategies for Indigenous Peoples [29]. That this policy is allowed to lie dormant is an example of how bureaucracy works and the banality of health inequity when it is normalized by society.”

For instance, how are you sure that when public health professionals become politicians, they will create a new pattern of government? 

Authors response: After interest groups, the collection of academics, researchers, and consultants is the next most important set of nongovernmental actors. Their impacts vary in a number of important ways. They affect alternatives more than agendas. Studies show that agendas are set by forces and actors outside the researcher-analyst community. Politicians turn to the researcher-analyst for proposals that are relevant to their concerns. The specific nature of a politician’s concerns, therefore, will reflect the individual politician’s interests. If the politician is a public health professional interested in health equity then they will use the power of their office to create those new patterns of government. See Kingdon, JW Agendas, Alternatives, and Public Policies (2nd Edition), Longman, New York, pp 253.

Reviewer 4 Report

37 "plan for the"

44: "They were also" Vs "there were also"?

48: Geographic Information System (GIS)

73: or advisory VS and advisory

95: policy-making model framework: 

104: Participating in the problem-identification and problem-solving streams revolve 

106: Whereas the policy stream focuses on processes that put problems on the political agenda 

pg4. table 1.

Not clear just by looking at the table in isolation what the levels of importance pertain to.

Slightly important Vs somewhat important? Are these being used interchangeably? Maybe just pick one and be consistent. 

pg5

4: ladder (political passivism)

10: and to attempt to

14: by through an emailed

16-18 : were the gatekeepers selected because they "call attention"? Maybe this can be worded differently to be clearer.

34: The documented annual release of 44-87 tons of mercury used in 

 pg6.

51: The level of importance of our petition to staffers, ... to the following criteria;

pg 7.

98: somewhat important or slightly important (i.e line 71 pg 6)

99 ...bank (IADB), the

98:102: should you make the 5 gatekeepers and their abbreviations consistent with the names used in subsequent paragraphs e.g. on pg 8 167, pg 9 229 etc? eg you seem to have used CRS instead of ORDA.

109: (SLMP)

116:Initiative for the Integration of the Regional Infrastructure of South America (IIRSA

116: should you clarify what the IIRSA actually is?

pg8:

189 - 195:pasionate but comes across somewhat overly preachy and emotional as opposed to factual, may put off some readers.

pg 10

252: ?Slightly important

290: if you to use "despite", maybe add that "its suggestion that ... remains relevant".

pg12.

371: lesson 6: is it worth addressing the 3rd rung also? (this suggestion applies to the discussion section also).

Author Response

Please note. All comments were corrected as indicated below then other corrections were made, paragraphs and sections revised which may have change the response and in some cases eliminated the text in question and the response.

1) 37 "plan for the"

Authors: Correction made

2) 44: "They were also" Vs "there were also"?

Authors: Correction made

3) 48: Geographic Information System(GIS)

Authors: Correction made

4) 73: or advisory VS and advisory

Authors: Correction made

5) 95: policy-making model framework:

Authors: Correction made

6) 104: Participating inthe problem-identification and problem-solving streams revolve

Authors: Correction made

7) 106: Whereas the policy stream focuses on processes that put problems on the political agenda

Authors: Correction made 

8) table 1. Not clear just by looking at the table in isolation what the levels of importance pertain to [and] Slightly important Vs somewhat important? Are these being used interchangeably? Maybe just pick one and be consistent. 

Authors: Revised categories to an “Index of Importance”, assigned criteria to score index, replaced four original categories with categories that have index levels scored 0 – 3.

9) 4: ladder (political passivism)

Authors: Revision made

10) 10: andto attempt to

Authors: Correction made

11) 14: bythrough an emailed

Authors: Correction made

12) 16-18 : were the gatekeepers selected because they "call attention"? Maybe this can be worded differently to be clearer.

Authors: Sentence was revised to read, “Gatekeepers, including staff persons, appointees, career bureaucrats, and nongovernmental actors, were selected to call attention to the structural causes of public health problems and possible government solutions”.

13) 34: The documented annual release of 44-87 tons of mercury used in 

Authors: Correction made

14) 51: The level of importance of our petition to staffers, ... to the following criteria;

Authors: Entire section (51-94) was revised to support and reflect changes made to Table 1 clarifying the categories and making them more quantitative.

15) pg 7. 98: somewhat important or slightly important (i.e line 71 pg 6) and 99 ...bank (IADB), the

Authors: Sentence 102 and 103 revised to read, “Only five discussed our proposed community health agenda and met the criteria for Index of Importance Level 3.”

16) 98:102: should you make the 5 gatekeepers and their abbreviations consistent with the names used in subsequent paragraphs e.g. on pg 8 167, pg 9 229 etc? eg you seem to have used CRS instead of ORDA.

Authors: CSR and ODRA consolidated under DOJ to reconcile what must be overlapping responsibilities.

17) 109: (SLMP)

Authors: Acronym added after name

18) 116:Initiative for the Integration of the Regional Infrastructure of South America (IIRSA

Authors: Name and acronym given  

19) 116: should you clarify what the IIRSA actually is?

Authors: Sentence revised to read, “The IADB did not accept the premise of our proposal, which was that from a public health perspective the assimilation model inherent in Initiative for the Integration of the Regional Infrastructure of South America (IIRSA) and the SLMP, which are development projects linking South America's economies to North American and Europe through new transportation, energy, and telecommunications projects, causes indigenous people to become disassociated, impoverished and alienated minorities, which in turn causes their health status to decline to unacceptable lows when measured in terms of death, disease, disability and the burgeoning rate of suicide [1-7, 27]

20) pg8: 189 - 195:pasionate but comes across somewhat overly preachy and emotional as opposed to factual, may put off some readers.

Authors: Paragraph was deleted

21) pg 10: 252: ?Slightly important

Authors: Sentence revised to “Beyond the five gatekeepers discussed above, there were other specialists who initially acknowledged and were willing to discuss our proposed public health agenda. Fourteen other gatekeepers agreed to meet but limited their interactions to interrogatories designed to learn facts while holding back information of their own, giving misinformation, or promising help that never materialized.”

22) 290: if you to use "despite", maybe add that "its suggestion that ... remains relevant".

Authors: Revision made.

23) 371: lesson 6: is it worth addressing the 3rd rung also? (this suggestion applies to the discussion section also).

Authors: Lesson 6 was revised to read, “Public health professionals can consider the third rung of Machenbach’s Imaginary Ladder of Political Activism by lobbying and by actively engaging politicians of specific political parties. While public health professionals may be subject to restrictions on the amount of “lobbying” in which they or their organization can engage, there are many activities they can engage in that can help educate policymakers and influence policy development that are not considered “lobbying” and therefore are not subject to limitations. The fourth rung is another option in which public health professionals become politicians themselves to obtain positions in government that would position them to create new policy commitments, create new patterns of governance, and include expectations and incentives that address health considerations in all high-level policy processes.”

Round 2

Reviewer 1 Report

Congratulations on how this work has emerged following the revisions.

The comprehensive history of the region now included is excellent, with appropriate degree of detail. It is well-written and precisely tailored to the needs of the reader of this article like myself with no previous knowledge of the region. Map is also of enormous value to me in understanding the geography of the work.

Likewise the information introduced on Mackenbach’s ladder greatly enhances the usefulness of this tool to the argument presented in the work.

Should this work contain a statement of ethical considerations? Has any institutional review board considered the work. I do not question the integrity of the work but the process which has led to its publication. As outsiders to the community the authors could reflect on their position in relation to the community for whom they are working.

Table 1: further refinements would make this more useful. For example there is repetition in the Indices of importance. These appear to be cumulative so it may not be necessary to repeat the full criteria each time. Further, I still struggle to interpret this information as it seems that from the data in the table there were no gatekeepers reaching level 3, while the text states that IADB, ICIM, ICRC,  U.S. DOJ, and Corporate Social Responsibility Executives from a Mining company reaching this level.

2.2- Focusing events: it would be useful to have citations for the references to the Tuskagee Syphilis and Baltimore lead-paint studies.

3. Results and discussion – line 270: “Fourteen candidate gatekeepers ostensibly agreed to meet to discuss our proposed global health agenda”- Could authors confirm that these are the entities in the Table who scored at least 1 in the index of importance? It may seem repetitious to authors but for me as a reader some duplicaiton of material would assist in confirming that I understand the points being made.

3.5 Engaged followership line 451: Milgrams’ work has been disputed, so it would be useful to note this even though the point is valid.

Would this work benefit from a statement of limitations, perhaps contributing to assessing how generalisation are the findings?

Conclusions are succinct, well-supported by the work and useful for practitioners in the area.

Author Response

1. Should this work contain a statement of ethical considerations? Has any institutional review board considered the work? I do not question the integrity of the work but the process which has led to its publication. As outsiders to the community the authors could reflect on their position in relation to the community for whom they are working.

Author’s Response: Ethics Approval - The primary intent of the work discussed in this paper was aimed at a specific public health problem, specifically to support indigenous people in the Guiana region with their efforts to self-diagnose public and environmental health problems concomitant with economic development projects and exposure to Hg contamination from mining. This paper addresses the secondary benefits of the community-led efforts at the interface between health, well-being and economic development. These public health intervention projects were reviewed by the University of Washington Institutional Review Board (IRB), Human Subjects Division (HSD). No reference number was assigned because the community-led project was conducted as a public health service and was deemed "non-research". Consequently, the work was not considered to be within the purview of institutional review. As such, these non-research investigations have yielded insights of generalizable value that merit dissemination, but the research versus non-research determination, which is based on the primary intent, remains unchanged.

2. Table 1: further refinements would make this more useful. For example, there is repetition in the Indices of importance. These appear to be cumulative so it may not be necessary to repeat the full criteria each time. Further, I still struggle to interpret this information as it seems that from the data in the table there were no gatekeepers reaching level 3, while the text states that IADB, ICIM, ICRC, U.S. DOJ, and Corporate Social Responsibility Executives from a Mining company reaching this level.

Authors Response: Table 1 has been deleted. It yielded little useful information, was more of a distraction than an enhancement to the narrative, and its absence does not seem to leave the manuscript wanting. The text is correct to say that the “IADB, ICIM, ICRC,  U.S. DOJ, and Corporate Social Responsibility Executives from a Mining company” reached the 3rd level.

3. Focusing events: it would be useful to have citations for the references to the Tuskagee Syphilis and Baltimore lead-paint studies.

3.1 Minamata Ref - Evolution of Our Understanding of Methylmercury as a Health Threat, Watanabe, C. and Satoh, H. Environmental Health Perspectives, Vol 104, Supplement 2, Aprl 1996.

3.2 Fukushima Daiichi Nuclear Power Plant (FDNPP) Accident Ref – On the divergences in assessment of environmental impacts from ionising radiation following the Fukushima accident, Strand, P., Sundell-Bergman, S., Brown, J.E., Dowdall, M., Journal of Environmental Radioactivity, 2017, 159e173, pp169-170.

3.3 Thalidomide Scandal Ref - Thalidomide and the Titanic: Reconstructing the Technology Tragedies of the Twentieth Century, George J. Annas, JD, MPH, and Sherman Elias, MD, American Journal of Public Health, 1999, Vol. 89(1): 98-101.

 3.4 Tuskegee Ref – Ethical Failures and History Lessons: The U.S. Public Health Service Research Studies  in Tuskegee and Guatemala, Reverby, S.M., Public Health Reviews (2107-6952). 2012, 34(1):1-18.

3.5 Baltimore Ref - Canaries in the mines: children, risk, non-therapeutic research, and justice. M Spriggs, J Med Ethics 2004;30:176–181.

4. Results and discussion – line 270: “Fourteen candidate gatekeepers ostensibly agreed to meet to discuss our proposed global health agenda”- Could authors confirm that these are the entities in the Table who scored at least 1 in the index of importance? It may seem repetitious to authors but for me as a reader some duplicaiton of material would assist in confirming that I understand the points being made.

Author’s Response: Deleting Table 1 should simplify the message, reduce the relate the text to the table and the potential for confusion.

5. Engaged followership line 451: Milgrams’ work has been disputed, so it would be useful to note this even though the point is valid.

Author’s Response: Despite the fact that Milgram’s work on followership and obedience to authority [42] has been widely criticized, the gatekeepers contacted all indicated they were committed to and constrained by mandates, policies, and the policy-making process and were not free to provide the academic and researcher-analyst community and leaders at the community level access (direct or indirect) to high level policy hearings, meetings, and advisory panels. There are two key reasons why this evidence could be viewed as consistent with claims that harmdoing is a product of engaged followership [Haslam, Reicher]. The first is that during our conversations with gate-keepers they felt compelled to defend themselves against being responsible for the health crises we described. The second is that the gatekeeper’s accounts of their actions revolved around their mandates, highest and best use of resources, and policies that contribute to the greater good, all expressions of trust in the high-level policy making process. We interpret these as manifestations of shared identity and engaged followership. 

6. Would this work benefit from a statement of limitations, perhaps contributing to assessing how generalisation are the findings?

Author’s Response: A ‘Statement of Limitations’ was added to the Methods Section by addressing three critical assumptions: Inherent in the objective  of this project were three critical assumptions: The first assumption was that public health practitioners, academics, researcher-analysts, and leaders at the community level, and other members of “Civil” society, could join government (public) and industry (private) as partners in PPPs to share the risks, responsibilities, and decision-making processes with the objective of collectively and more effectively addressing a common goal.

The second assumption was that communication channels between those inside and those outside of government are open and ideas and information pass through these channels in a network of involved people, somewhat independent of their formal positions.

The third assumption was that common values, orientations, and world views would form bridges between those inside and those outside of government.

Reviewer 2 Report

The article problems subsist. The fundamental problems have just been masked.

Author Response

The article problems subsist. The fundamental problems have just been masked.

Author’s Response: The primary intent of our work was to provide a public health service. This paper communicates the secondary benefits of our work by discussing the problems we encountered. Our hope was to unmask the obstacles we encountered that complicated our efforts to address the structural causes and social determinants of poor health and contribute to efforts to develop methods to more effectively address these very serious issues.